# PDPN Is Expressed in Various Types of Canine Tumors and Its Silencing Induces Apoptosis and Cell Cycle Arrest in Canine Malignant Melanoma

**DOI:** 10.3390/cells9051136

**Published:** 2020-05-05

**Authors:** Masahiro Shinada, Daiki Kato, Satoshi Kamoto, Sho Yoshimoto, Masaya Tsuboi, Ryohei Yoshitake, Shotaro Eto, Namiko Ikeda, Kohei Saeki, Yuko Hashimoto, Yosuke Takahashi, James Chambers, Kazuyuki Uchida, Mika K. Kaneko, Naoki Fujita, Ryohei Nishimura, Yukinari Kato, Takayuki Nakagawa

**Affiliations:** 1Laboratory of Veterinary Surgery, Graduate School of Agricultural and Life Sciences, The University of Tokyo, 1-1-1 Yayoi, Bunkyo-ku, Tokyo 113-8657, Japan; ms003_shina@yahoo.co.jp (M.S.); veteronut@gmail.com (S.K.); syoshimo@g.ecc.u-tokyo.ac.jp (S.Y.); g050446@gmail.com (R.Y.); shotaro.nyantaro@gmail.com (S.E.); tsubaki5228@gmail.com (N.I.); kohei.saeki.1987@gmail.com (K.S.); apom@mail.ecc.u-tokyo.ac.jp (N.F.); arn@mail.ecc.u-tokyo.ac.jp (R.N.); anakaga@mail.ecc.u-tokyo.ac.jp (T.N.); 2Veterinary Medical Center, The University of Tokyo, 1-1-1 Yayoi, Bunkyo-ku, Tokyo 113-8657, Japan; atsuboi@mail.ecc.u-tokyo.ac.jp (M.T.); yuko14hashi@gmail.com (Y.H.); yousuke1982_root_story@yahoo.co.jp (Y.T.); 3Laboratory of Veterinary Pathology, Graduate School of Agricultural and Life Sciences, The University of Tokyo, 1-1-1 Yayoi, Bunkyo-ku, Tokyo 113-8657, Japan; achamber@mail.ecc.u-tokyo.ac.jp (J.C.); auchidak@mail.ecc.u-tokyo.ac.jp (K.U.); 4Department of Antibody Drug Development, Tohoku University Graduate School of Medicine, 2-1 Seiryo-machi, Aoba-ku, Sendai 980-8575, Japanyukinarikato@med.tohoku.ac.jp (Y.K.); 5New Industry Creation Hatchery Center, Tohoku University, 2-1 Seiryo-machi, Aoba-ku, Sendai 980-8575, Japan

**Keywords:** podoplanin, cancer, Ki67, melanoma, squamous cell carcinoma, apoptosis, cell cycle

## Abstract

Podoplanin (PDPN), a small transmembrane mucin-like glycoprotein, is ectopically expressed. It is also known to be linked with several aspects of tumor malignancy in some types of human tumors, including invasion, metastasis, and cancer stemness. However, there are few reports on the expression of dog PDPN (dPDPN) in canine tumors, and the association between dPDPN and tumor malignancy has not been elucidated. We identified that 11 out of 18 types of canine tumors expressed dPDPN. Furthermore, 80% of canine malignant melanoma (MM), squamous cell carcinoma, and meningioma expressed dPDPN. Moreover, the expression density of dPDPN was positively associated with the expression of the Ki67 proliferation marker. The silencing of dPDPN by siRNAs resulted in the suppression of cell migration, invasion, stem cell-like characteristics, and cell viability in canine MM cell lines. The suppression of cell viability was caused by the induction of apoptosis and G2/M phase cell cycle arrest. Overall, this study demonstrates that dPDPN is expressed in various types of canine tumors and that dPDPN silencing suppresses cell viability through apoptosis and cell cycle arrest, thus providing a novel biological role for PDPN in tumor progression.

## 1. Introduction

Podoplanin (PDPN), also known as PA2.26, gp38, T1α, and Aggrus, is a small transmembrane mucin-like glycoprotein [1,2,3]. PDPN is expressed in various types of normal cells such as renal podocytes, pulmonary type I alveolar cells, lymphatic endothelial cells (LECs), osteocytes, mesothelial cells, glia cells, some types of neurons, and fibroblasts [1,2,3]. PDPN has been reported to have various roles, which are dependent on cell type and the timing of its expression [2]. In the embryo, PDPN expression plays a crucial role in organogenesis by regulating epithelial–mesenchymal transition (EMT) [2]. Indeed, PDPN null mice die shortly after birth because of the malfunction of alveoli and heart hypoplasia [2,4,5]. In adult human epidermal keratinocytes and fibroblasts, PDPN promotes cell motility through the modification of actin cytoskeleton organization [6,7]. Meanwhile, in adult immune cells, PDPN regulates the immune system and the inflammatory microenvironment by regulating various types of cytokines [8,9].

Moreover, PDPN has been reported to be overexpressed in various types of human tumors, and it is reported that 100% of glioma [10], 100% of lymphangioma [11], 98% of germ cell tumor [12], 94% of angiosarcoma [13], 86% of malignant mesothelioma [14], 80% of squamous cell carcinoma (SCC) [15,16], 80% of osteosarcoma [17], 78% of hemangioendothelioma [18], 36% of pleomorphic carcinoma of the lung overexpressed PDPN [19]. Some reports demonstrated that PDPN is associated with tumor malignancy through the promotion of tumor cell invasion and distant metastasis with or without platelet aggregation [1,2]. In human SCC, PDPN promotes tumor cell migration and invasion by cytoskeleton remodeling via Rho signaling [20]. In human SCC and malignant pleural mesothelioma cell lines, PDPN induces EMT and promotes tumor cell invasion and distant metastasis by binding with platelets [21]. Furthermore, anti-PDPN antibodies, which neutralize PDPN–platelet interaction, suppress the tumor growth and pulmonary metastasis of human melanoma and mesothelioma in a mouse model [22,23]. Therefore, it is considered that PDPN enhances tumor malignancy and thus could be used as a therapeutic target for some types of tumors.

In a similar manner to humans, PDPN has been reported to be expressed in renal podocytes, alveolar epithelial cells and LECs in dogs [24,25,26]. However, there are few reports on the expression of dog PDPN (dPDPN) in canine tumors. Spontaneously occurring canine cancer is the most common cause of death in dogs, as well as humans. Approximately one in three dogs will be diagnosed with cancer during their lifetime, and cancer currently accounts for about half of the deaths of all dogs older than 10 years [27,28,29]. Since dogs live in close proximity to humans, they are influenced by similar environmental factors which can lead to cancer development [27,28,29]. Furthermore, many features of spontaneously occurring canine cancers are similar to those of human cancers in terms of histological morphology, biological behavior, molecular mechanisms and response to conventional therapy [27,28,29]. Although we have reported the expression of dPDPN in both normal tissues and tumor cells, including canine SCC [25,30], no research has provided a comprehensive profile of dPDPN expression in different types of canine tumors and the role of dPDPN in canine tumors remains to be elucidated. The aim of this study was to investigate the expression of dPDPN in various types of canine tumors and to investigate its role in tumor progression.

## 2. Materials and Methods

### 2.1. Specimens

A total of 159 paraffin-embedded tumor tissues were evaluated. Tumor tissues surgically removed from dogs between 2011 and 2017 at the Veterinary Medical Center, University of Tokyo, were included in this study. Permission for resected tissue collection and usage for this study was obtained from the dogs’ owners. Tissue samples were diagnosed by veterinary pathologists certified by the Japanese College of Veterinary Pathologists at the Department of Veterinary Pathology at the University of Tokyo.

### 2.2. Immunohistochemistry of dPDPN and Ki67

Archived formalin fixed paraffin embedded (FFPE) tissues were obtained and sectioned at 4 µm thickness. Sections were dewaxed and rehydrated in xylene and graded ethanol. Antigen retrieval was performed using Tris- ethylenediaminetetraacetic acid (EDTA) buffer pH 9.0 or citrate buffer pH 6.0 at 121 °C for 10 min. After antigen retrieval, sections were washed with Tris-buffered saline with 0.1% Tween-20 (TBS-t). Endogenous peroxidase was blocked by 3% hydrogen peroxide in deionized water for 30 min. Specimens were incubated for 1 h in TBS-t with 10% skimmed milk for dPDPN or 5% normal goat serum for Ki67 at room temperature before incubation with primary antibodies. The primary antibodies used are summarized in Appendix A. After washing, sections were incubated with a horseradish peroxidase (HRP)-conjugated anti-mouse antibody (Envision+ System-HRP Labelled Polymer; K4001; Agilent Technologies, Santa Clara, CA, USA) for 30 min at room temperature. Sections were then washed with TBS-t, incubated with 3,3′-diaminobenzidine (Dojindo Laboratories, Rockville, MD, USA) solution for 2 min and counterstained with Mayer’s hematoxylin for 50 s.

### 2.3. Evaluation of dPDPN and Ki67 Expression in Tumor Tissues

Immunostained sections were examined under a light microscope. Tissue samples were detected as tumor region and stroma region and referenced by hematoxylin and eosin (HE) staining. Only the number of positively stained tumor cells in the tumor region was counted, and specimens were considered to be positive for dPDPN if >10% of the tumor cell was stained at a moderate to strong intensity in five independent high-power (400×) fields. Canine normal lung tissue was used as the positive control for dPDPN staining.

To calculate the percentage of Ki67-positive tumors cells in tumor tissues, three 400× fields within areas with the heaviest Ki67 stains were manually assessed. Only tumor cells with nuclear staining were counted. The number of Ki67-stained tumor cells in the field was counted, and the percentage of Ki67-stained tumor cells was calculated by dividing the number of Ki67-stained tumor cells by the number of all tumor cells in the area.

To investigate the association between PDPN expression and Ki67 expression, canine malignant melanoma tissues (n = 10) were divided into two groups, PDPN high-expression group (n = 5) and PDPN low-expression group (n = 5), according to the intensity of dPDPN staining. The percentage of Ki67-stained cells in each group was compared.

### 2.4. Cell Lines and Cell Culture

Nineteen cell lines were used in this study and specific details of each cell line are summarized in Appendix A. Cells were cultured in RPMI-1640 (FUJIFILM Wako Pure Chemical Corporation, Osaka, Japan) supplemented with 10% heat-inactivated fetal bovine serum (FBS; Thermo Fisher Scientific Inc., Waltham, MA, USA) and 5 mg/L gentamicin (Sigma-Aldrich Corp., St. Louis, MO, USA) or in DMEM/Ham’s F-12 medium (FUJIFILM Wako Pure Chemical Corporation) with 10% FBS and 100 U/mL penicillin, 100 µg/mL streptomycin (FUJIFILM Wako Pure Chemical Corporation), depending on the cell line (Appendix A). Cells were maintained at 37 °C in a humidified atmosphere with 5% CO_2_.

### 2.5. Quantification of dPDPN mRNA Expression

Total RNA was extracted from each cell line in exponential growth phase using an RNA extraction reagent (TRI Reagent, Cosmo Bio, Tokyo, Japan) and reverse-transcribed using reverse transcriptase (ReverTra Ace, Toyobo, Osaka, Japan) according to manufacturer’s instructions. Real-time PCR was performed using a premixed reagent (THUNDERBIRD SYBR qPCR Mix, Toyobo), specific primers *PDPN* (forward; 5’-CCAGAGAGAAAGTAGGTGAAGAC-3’, reverse; 5’-AAATGTGTTGGTAGAAGGGCA-3’), and a real-time PCR system (StepOnePlus, Thermo Fisher Scientific, Inc.). The qPCR conditions were as follows: initial denaturation at 95 °C for 10 min, then 40 cycles of denaturation at 95 °C for 15 s, annealing and elongation at 60 °C for 60 s. All samples were analyzed in triplicate. Gene expression was calculated using the ΔΔCt method. Expression values were normalized using an internal control, *GAPDH* (forward, 5’-TGACACCCACTCTTCCACCTTC-3’, reverse, 5’-CGGTTGCTGTAGCCAAATTCA-3’).

### 2.6. Flow Cytometry

Washed cells were resuspended in PBS with 5% FBS and 0.01% sodium azide (FACS buffer). Cells were pelleted by centrifugation at 500× *g* for 3 min. After washing the cells three times with FACS buffer, cells were incubated with specific antibodies for dPDPN (mouse monoclonal, clone: PMab-38, ZENOAQ Resource, Fukushima, Japan [26,31]) for 30 min on ice. After washing three times, cells were incubated with Alexa Fluor 488 anti-mouse IgG antibody (Abcam, Cambridge, England, UK) for 30 min on ice in the dark. All flow cytometric analyses were performed with BD FACSverse (BD, Franklin Lakes, NJ, USA) and data were analyzed using BD FACSuite software (BD, ver 8.0).

### 2.7. dPDPN Knockdown by Small Interfering RNA

Target gene-specific and control small-interfering RNA (siRNA) were purchased from Sigma-Aldrich Corp. Target sequences for dPDPN were as follows; siRNA#1: 5’-GAGAGUGUAACAGACUUAC-3’, siRNA#2: 5′-AGGAUGGGCCGACUCAAGA-3′. Mi and CMM12 were seeded at a density of 7.9 × 10^2^ cells/cm^2^ and 2.6 × 10^3^ cells/cm^2^, respectively. After incubation for 24 h, Mi and CMM12 were incubated with 20 nM or 50 nM siRNAs and 2 or 4 µg/mL Lipofectamine^TM^ RNAiMAX (Thermo Fisher Scientific, Inc.) in Opti-MEM (Thermo Fisher Scientific, Inc.) and each growth medium with 10% FBS, respectively. After incubation for 8 h, Mi medium was removed and fresh medium was added. As a negative siControl, MISSION^®^ siRNA Universal Negative Control (Mission SIC 001, Sigma-Aldrich Corp.) was used. siRNA-transfected cells were incubated at 37 °C in 5% CO_2_ until the assay was carried out.

### 2.8. Transwell Migration/Invasion Assay

Culture inserts (24-well permeable support, 8.0 µm pore, Corning, Corning, NY, USA) were set on a 24-well companion plate (Corning). For the migration assay, uncoated inserts were used, and for the invasion assay, inserts were incubated with 200 µL Matrigel (200 µg/mL) (BD) for 3 h at 37 °C before using. After preparing culture inserts, a cell suspension containing 1.0 or 2.0 × 10^4^ cells in 400 µL serum-free medium was added to each culture insert. Medium with 10% FBS was added to the lower chamber of the companion plate as a chemoattractant. Plates were then incubated for another 24 h at 37 °C in a humidified 5% CO_2_ atmosphere. Cells were fixed and stained with PBS containing 6% glutaraldehyde and 0.5% crystal violet and images of each culture insert were captured under magnification (×200). Three images per one culture insert were randomly captured, and all cells in each image were manually counted as migrated/invaded cells.

### 2.9. Sphere Forming Assay

Cells were plated as single cell suspensions in 24-well ultra-low attachment plates at 500 cells/mL density to obtain single cell-derived tumor spheres after siRNA treatment for 48 h. Cells were grown in DMEM/F-12 medium, 20 ng/mL epidermal growth factor (Sigma-Aldrich Corp.), 20 ng/mL basic fibroblast growth factor (FUJIFILM Wako Pure Chemical Corporation), B27 supplement (Thermo Fisher Scientific Inc.) and 5 mg/L gentamicin (Sigma-Aldrich Corp.). Spheres with a diameter >100 µm were counted after 3 days for the Mi cell line and 5 days for the CMM12 cell line.

### 2.10. Cell Proliferation Assay

After a 48-h incubation for siRNA transfection, cells that were not stained with trypan blue (Sigma-Aldrich Corp.) were counted as live cells.

### 2.11. Cell Cycle Analysis

Mi and CMM12 cell lines were incubated with siRNAs for 48 h and 72 h, respectively. Subsequently, the cells were trypsinized and fixed with 100% ice-cold ethanol for 20 min on ice. After washing with PBS, cells were stained with 50 µg/mL propidium iodide (Sigma-Aldrich Corp.), 0.1 mg/mL RNase A (Roche Diagnostics, Basel, Switzerland) and 0.05% Triton-X 100 (Sigma-Aldrich Corp.) for 40 min at 37 °C. Stained cells were immediately analyzed by flow cytometry.

### 2.12. Western Blotting Analysis

Cells were lysed using RIPA buffer (50 mM Tris-HCl, 150 mM NaCl, 5 M EDTA, 1% Triton-X, and 0.1% sodium dodecyl sulfate (SDS)) supplemented with 10 mM NaF, 2 mM Na_3_VO_4_, and a complete Mini Protease Inhibitor Cocktail (Roche Diagnostics). Protein content was measured using a BCA Protein Assay Kit (Thermo Fisher Scientific Inc.). Proteins were resolved by SDS–polyacrylamide gel electrophoresis and transferred onto polyvinylidene difluoride membranes (Bio-Rad Laboratories, Hercules, CA, USA). Membranes were blocked in TBS-t containing 5% skimmed milk for 1 h at room temperature before incubating with the primary antibodies overnight at 4 °C. The primary antibodies used and their working conditions are summarized in Appendix A. Membranes were then incubated with horseradish peroxidase-conjugated anti-mouse or anti-rabbit IgG antibody (GE Healthcare, Buckinghamshire, England, UK; 1:10,000, TBSt-milk, room temperature) as the secondary antibody for 1 h. The membranes were developed using the ECL Prime Western Blotting Detection System (GE Healthcare) and luminescence was captured and quantified using an imaging system (ChemiDoc Image Lab, Bio-Rad Laboratories).

### 2.13. Detection of Apoptotic Cells

To detect apoptotic tumor cells, siRNA-treated cells were stained using the ApoAlert Annexin V-FITC Apoptosis Kit (Takara Bio Inc, Shiga, Japan) according to the manufacturer’s instructions. The staining pattern was analyzed by flow cytometry.

### 2.14. Statistical Analysis

All data were shown as the mean ± standard deviation (SD). Welch’s t-test was used to compare differences between two sample groups. The Dunnett test was used to compare differences among three sample groups. All analysis was performed using R software (ver. 3.6.1, R Development Core Team, 2019) and the multcomp R package (ver. 1.4-12, Hothorn et al., 2019). Values of *p* < 0.05 were considered statistically significant.

## 3. Results

### 3.1. dPDPN Was Expressed In Canine Tumor Tissues and Cell Lines

Immunohistochemical analysis for dPDPN expression was performed in 159 specimens from 18 types of canine tumors. The positive staining ratio for tumor cells varied among tumor types (Table 1). High frequency of dPDPN expression was observed in canine tumor tissues of malignant melanoma (MM: 8/10; 80%), SCC (8/10; 80%), and meningioma (4/5; 80%). All canine MM tissues were derived from oral cavity, and canine SCC tissues were derived from head and neck (n = 9) and skin (n = 1). 

Moderate frequency of dPDPN expression was observed in lung adenocarcinoma (4/10; 40%), mast cell tumor (4/10; 40%), fibrosarcoma (3/10; 30%), and prostate cancer (1/4; 25%). Low frequency of dPDPN expression was observed in osteosarcoma (2/10; 20%), mammary carcinoma (2/10; 20%), peripheral nerve sheath tumor (1/10; 10%), and anal sac adenocarcinoma (1/10; 10%). No expression of dPDPN was observed in transitional cell carcinoma (TCC), thymoma, thyroid cancer, hepatic cell carcinoma, hemangiosarcoma, ceruminous carcinoma, and intestinal adenocarcinoma (Table 1 and Figure 1A). More than 50% of tumor cells were stained in all canine MM tissues. More than 50% of the tumor cells were stained in about half of SCC and meningioma tissues. In all other tumor types, the proportion of dPDPN positive tumor cells varied from 10% to 40%.

The membranes of tumor cells were strongly stained (Figure 1A). Localization of dPDPN to the cell membrane was observed in all types of tumors, and there were no clear differences in the intensity of the membrane staining among tumor types. The staining patterns of the positively stained tumor tissues were varied, including diffuse and focal patterns. We could not find a specific staining pattern according to tumor type.

Because dPDPN was frequently expressed in canine MM tissues, we investigated the association between dPDPN expression and canine MM malignancy. We investigated the association between dPDPN expression and Ki67, the proliferation marker, which is reported to be a poor prognostic marker of canine MM. The percentage of Ki67-stained tumor cells in tissues with high expression of dPDPN was significantly higher than in tissues with low expression of dPDPN (mean; 46.1% vs. 23.0%, *p* = 0.020) (Figure 1B).

Based on the results of immunohistochemical evaluation, dPDPN is overexpressed on the surface of cell membranes in many types of canine tumors. To evaluate the function of dPDPN in canine tumors, the expression of dPDPN was evaluated by qPCR analysis in 17 canine cell lines, which included four types of canine tumor cell lines. In a similar manner to the immunohistochemical findings, *dPDPN* mRNA was expressed in canine MM cell lines only, and not in TCC, mammary carcinoma and osteosarcoma cell lines (Figure 1C). Furthermore, protein expression of dPDPN in canine MM cell lines was evaluated by flow cytometry analysis, because dPDPN protein expression could not be detected by Western blot analysis using our antibodies. Surface dPDPN protein expression was observed in Mi and CMM12 cell lines, whereas we could not detect dPDPN protein expression in LMeC and Pu cell lines (Figure 1D).

### 3.2. Cell Viability of Canine Malignant Melanoma Cell Lines Was Reduced by dPDPN Knockdown

To investigate the role of dPDPN in several aspects of canine MM malignancy, such as tumor cell migration, invasion, cancer stemness and cell proliferation, a dPDPN knockdown experiment was performed in canine MM cell lines (Mi and CMM12) using siRNAs. Treatment with dPDPN-specific siRNA#1 and #2 significantly inhibited dPDPN expression in both Mi and CMM12 cell lines, compared to the cell lines treated with control-scrambled siRNA (Figure 2A). Knockdown of dPDPN resulted in a significant reduction in the migration capacity of the Mi cell line. A slight reduction in the migration capacity of the CMM12 cell line was also observed, although it was not significant (Figure 2B). Knockdown of dPDPN expression also resulted in the significant inhibition of the invasion capacity of Mi cell line. We could not, however, evaluate the invasion capacity of the CMM12 cell line, because cells did not invade the Matrigel during the evaluated period of 24 h (Figure 2C).

To investigate the association between dPDPN and cancer stemness, a sphere-forming assay was performed. The number of spheres developed from a single cell of the Mi cell line was significantly reduced following dPDPN-specific siRNAs treatment. On the contrary, dPDPN knockdown in CMM12 cell line did not reduce the number of spheres (Figure 2D).

Finally, the association between dPDPN and cell viability was also investigated. Treatment with dPDPN-specific siRNAs significantly suppressed cell viability in both Mi and CMM12 cell lines after 48 h, with the number of living cells being reduced by 40% to 80% (Figure 2E).

### 3.3. Tumor Cell Apoptosis Induced by dPDPN Knockdown

To investigate the mechanism of inhibition of cell viability by dPDPN knockdown, the effect of dPDPN knockdown on the induction of apoptosis was evaluated. After 48 h of dPDPN-specific siRNAs treatment for both Mi and CMM12 cell lines, the percentage of Annexin V-positive tumor cells was significantly increased (Figure 3A,B). Next, we used Western blot to evaluate the expression of Cleaved Caspase 3 protein, an indicator for initiation of apoptosis, in siRNA-treated Mi cell line. We showed that Cleaved Caspase 3 protein expression was increased in the dPDPN-knocked-down Mi cell line (Figure 3C). These results demonstrated that dPDPN knockdown induced the apoptosis of tumor cells in canine MM cell lines.

### 3.4. dPDPN Knockdown Induced Cell Cycle Arrest at G2/M Phase

To further investigate the mechanism of cell viability suppression caused as a result of dPDPN knockdown, we investigated the effect of dPDPN knockdown on the cell cycle. In the Mi cell line, flow cytometry analysis demonstrated that the ratio of G0/G1-phase cells was significantly decreased and the ratio of G2/M-phase cells was significantly increased in siRNA-treated cells compared to the control siRNA treated cells. In siRNA-treated CMM12 cell line, the ratio of S-phase cells was significantly decreased and the ratio of G2/M-phase cells was significantly increased compared to the control siRNA treated cells. (Figure 4A,B). The increase in G2/M-phase cells in both Mi and CMM12 cell lines indicated that G2/M cell cycle arrest could be induced by siRNAs-treatment. We next used Western blot to evaluate the expression of G2/M cell cycle related proteins in the siRNA-treated Mi cell line, which exhibited extensive G2/M cell cycle arrest. The expression of phospho-cdc2 (Tyr15), phospho-cell checkpoint kinase 2 (p-Chk2) (Thr68), p21 and phospho-p53 (Ser15) was upregulated in the dPDPN-specific siRNAs treated cell line (Figure 4C). Moreover, the expression of phospho-ataxia telangiectasia-mutated (p-ATM) (Ser1981), one of the regulators of Chk2 and p53 cascade, was upregulated in the dPDPN-specific siRNA-treated cell line (Figure 4D). Overall, the data suggests that dPDPN knockdown induced G2/M cell cycle arrest by altering G2/M-corresponding proteins in canine MM cell lines.

## 4. Discussion

In the present study, we firstly performed a comprehensive analysis of dPDPN expression in various types of canine tumors. We demonstrated that dPDPN was expressed in many types of canine tumors. These results indicate the importance of exploring the link between dPDPN and the tumorigenesis and/or malignancy of canine tumors. During the evaluation of clinical samples, we surprisingly found that dPDPN is significantly associated with the cell viability of canine MM cells, which has not been reported in previous papers, even in other species. Following these findings, we investigated the molecular function of dPDPN in canine MM cell lines. We demonstrated that dPDPN regulates both cell apoptosis and the G2/M phase of the cell cycle.

Immunohistochemical analysis revealed that 11/18 types of canine tumors expressed dPDPN, and that 80% of canine MM, SCC and meningioma expressed dPDPN. The positive ratios of dPDPN in canine tumors were similar to those observed in homogenous human tumors. Indeed, human PDPN expression has been observed in 70% of human cutaneous MM [32,33], 70–80% of SCC [15,16], and 90% of meningioma [34]. 

In this study, 80% of canine MM patients expressed dPDPN, and dPDPN was broadly expressed in canine MM tissues. Canine and human MM derives from various organs including oral cavity, skin, ocular, ungual, and digit [35,36,37,38]. Among these, oral MM is the most common type of canine MM and cutaneous melanoma is the most common type of human melanoma [35,36,37,38]. Although oral melanoma is classified as mucosal-type melanoma and is different from cutaneous type melanoma, PDPN was frequently expressed in both canine oral MM and human cutaneous MM, which might indicate a similarity in the mechanism for aberrant PDPN expression in these MMs. Canine MM has highly aggressive characteristics and poor prognosis. Generally, complete surgical removal is performed for the management of the local tumor, but it frequently fails. Local tumor recurrence rates have been reported to range from 22–48% because of the rapid cell proliferation and the high invasiveness of canine MM [39,40,41,42,43]. The median survival time after surgery has been reported to range from 150 to 318 days, with the one-year survival rate being less than 35% [39,40,42,43]. This is due to a high metastatic rate of more than 70% [44]. To investigate the association between dPDPN expression and the tumor malignancy of canine MM, we evaluated the link between dPDPN expression and Ki67. Importantly, MM canines with a high expression of dPDPN showed a significantly higher percentage of Ki67-positive tumor cells than those with a low expression of dPDPN. Ki67, which is a marker of cell proliferation, has been reportedly related to rapid tumor progression, increased vascular invasion, and distant metastasis in canine and human MM [37,45,46]. It has also been reported that higher expression of Ki67 in canine MM is linked to a significantly shorter prognosis. Similarly, the four-year survival rate in canine MM with a high expression of Ki67 was 11%, while that in canine MM with low Ki67 was 83% [45]. Therefore, high PDPN expression might be a poor prognostic factor in canine MM. Furthermore, dPDPN is potentially related to tumor malignancy in canine MM. Following these findings, we used an in vitro assay to evaluate the role of dPDPN in canine MM.

In a similar manner to previous reports [15,16], we found dPDPN to be expressed in 80% of canine SCC. Head and neck SCC is common among dogs and is known to have a poor prognosis due to its high invasive and metastatic characteristics [47,48,49]. In a similar manner to canine SCC, head and neck SCC is also common in humans and is known to have highly invasive and metastatic characteristics [32,50,51]. In a previous report, about 50% of people with SCC developed nodal metastases and died [52,53]. In human SCC, PDPN expression has been related with increased tumor progression and metastasis rates [51]. In addition, PDPN has been shown to promote the invasion and growth of human SCC cell lines by inducing cytoskeleton remodeling and platelet aggregation [20,21,54]. Based on this, PDPN-neutralizing antibodies have been developed and, in fact, such antibodies have been shown to suppress the tumor progression of human SCC in a mouse model [54]. Overall, these reports indicate that dPDPN expression in canine SCC is potentially linked to tumor malignancy via effects on tumor migration, invasion, and metastasis, as described in human SCC. Thus, dPDPN is a potential therapeutic target in canine SCC. It is important to note that, in this study, we could not find an association between dPDPN expression and clinicopathological features, including survival time of canine SCC, due to the small sample size. A further investigation to clarify the clinical significance and molecular role of dPDPN in canine SCC would be of interest.

In this study, dPDPN expression was also observed in tumor cells of canine fibrosarcoma, osteosarcoma, lung adenocarcinoma, mast cell tumor, prostate tumor, mammary carcinoma, peripheral nerve sheath tumor, and anal sac adenocarcinoma. There were some similarities and some discrepancies in the expression of PDPN between canine and homogenous human tumors. For example, PDPN was expressed in human fibrosarcoma cell lines and human osteosarcoma tissues, which are rare tumors in humans but are frequently observed tumors in dogs [17,55]. On the other hand, human lung adenocarcinoma cells have been reported to not express PDPN [12,17]. Thus, further research into the role of PDPN in tumor biology is necessary. In addition, there are no reports on the expression of PDPN in human mast cell tumors and peripheral nerve sheath tumors, which are rare in humans. The occurrence rates of each type of tumor in dogs and humans are different, with some tumors that are rare in humans being common in dogs (e.g., fibrosarcoma, osteosarcoma, mast cell tumor, and peripheral nerve sheath tumor). However, given that PDPN is commonly expressed in certain types of human and dog tumors, canines could be used as models for the study of PDPN function in such tumors. 

As mentioned above, canine MM has highly invasive and metastatic characteristics, which can often lead to death. Furthermore, there have been reports that the blocking of PDPN suppresses pulmonary metastasis of human MM in a mouse model [22]. In addition, most canine MM is resistant to chemotherapy and radiation therapy [35,36,56]. Many reports have suggested that cancer stemness is responsible for chemo- and radio-resistance, and in turn, PDPN has been shown to be related to cancer stemness in human SCC [57]. Based on this research, we evaluated the link of dPDPN with the invasion, metastasis and stemness abilities of canine MM cell lines. Results showed that knockdown of dPDPN in Mi cells resulted in the suppression of migration, invasion capacity, and stem cell-like characteristics. However, we did not observe a significant suppression of cell migration, invasion, and stem cell characteristics in the CMM12 cell line. This might be explained by the fact that the CMM12 cell line has a lower capacity for migration, invasion and sphere formation than the Mi cell line. It was possible that the reduction in capacity induced by knockdown was relatively small and the change was not detectable. Although the concordance phenomenon was not observed between Mi and CMM12 cell lines, possible relationship between dPDPN and migration, invasion, and cancer stemness in canine MM was demonstrated. It has been reported that PDPN regulates cell migration, invasion and stemness via signaling through the Rho family, which are regulators of cellular motility and adhesion through cytoskeleton remodeling [7,20,58,59,60,61]. The mechanism by which signaling via Rho proteins is regulated by PDPN varies depending on cell types. Previous reports of human SCC demonstrated that PDPN promotes cell migration and invasion by suppressing RhoA activity [20]. On the other hand, PDPN expression in human fibroblast cells and Madin–Darby canine kidney cells promotes cell migration and invasion by activating RhoA signaling [7,58]. In another report, PDPN was shown to enhance sphere-forming capacity in the human SCC cell line by activating Rho signaling [62]. Because it is reported that gene expression related to Rho family signaling is enriched in canine MM [63], it might be possible that dPDPN promotes cell migration, invasion and stemness by activating Rho family signals. A further investigation to clarify the specific mechanism by which dPDPN induces migration, invasion and stemness is necessary.

We found that PDPN knockdown significantly suppressed cell viability in both Mi and CMM12 cell lines. Thus, apoptosis and cell cycle analysis were performed in dPDPN-specific siRNA-treated cell lines. The results showed that dPDPN knockdown increased Annexin V-positive cells in both Mi and CMM12, as well as increased the protein expression of Cleaved Caspase 3. Positive Annexin V staining indicates that cells have been induced to undergo apoptosis, and Caspase 3, an effector of the initial death cascade, is a representative marker of a cell’s entry point into the apoptotic signaling pathway [64]. Based on these results, we suggest that dPDPN knockdown induces apoptosis in canine MM cell lines. However, because there are no published reports showing the association between PDPN and induction of apoptosis induction and there are few reports on downstream of PDPN, it was difficult to speculate on the specific mechanism by which dPDPN knockdown induces apoptosis. Based on a few previous findings, we speculated that some of the potential mechanisms related to our observations could be related to the mitogen-activated protein kinase (MAPK) pathway, nuclear factor-kappaB (NF-κB) pathway, and Rho-associated protein kinase (ROCK) pathway. Many studies have shown that the MAPK pathway plays a crucial role in cellular survival signaling in response to extracellular stimuli, leading to its resistance to extracellular stress-induced apoptosis [65,66]. In canine MM, the activation of MAPK signals through several mechanisms have been reported [63,67,68]. Furthermore, it has been reported that PDPN activates MAPK signals in macrophages [69]. Thus, it may be possible that dPDPN expressed in canine MM cell lines activates MAPK signals and the knockdown of dPDPN-induced apoptosis in canine MM cell lines. There are also reports that NF-κB activation inhibits cell apoptosis and that NF-κB signals in macrophages were suppressed by PDPN inhibition [69,70]. Therefore, it is possible that dPDPN knockdown suppressed the NF-κB signaling pathway, thus resulting in the induction of apoptosis in canine MM cell lines. Many studies have shown that activation of the ROCK signal induces apoptosis, and the ROCK signal is downstream of Rho family proteins, which are regulated by PDPN [71,72]. It is possible that dPDPN regulates ROCK signaling via its interaction with Rho family proteins in canine MM cell lines and that dPDPN knockdown in canine MM induces apoptosis by activating the ROCK signal. In this study, we could not identify the specific mechanism of apoptosis regulated by dPDPN. The identification of the signaling pathway regulating PDPN-induced apoptosis would allow PDPN-induced apoptosis signaling to be used as a novel therapeutic target in various types of PDPN-expressing tumors.

In this study, we demonstrated that dPDPN knockdown in canine MM cell lines induced G2/M cell cycle arrest. To the best of our knowledge, this is the first report showing the association between PDPN and G2/M cell cycle arrest. G2/M cell cycle arrest is mainly induced by DNA damage. When DNA damage occurs, the Chk cascade and p53 cascades are activated, which, in turn, suppress the activation of Cyclin B1 and Cdc2 complex via Cdc25c and p21, respectively [73,74,75,76]. As a result, G2/M cell cycle arrest is induced. In this study, dPDPN knockdown increased the proportion of cells at G2/M phase, and upregulated the expression of phosphorylated Cdc2, which is an inactive form, phosphorylated Chk2, phosphorylated p53, and p21. Although we could not evaluate the phosphorylation of Cdc25c because of the lack of anti-canine phosphorylated Cdc25c/Cdc25c antibodies, these results suggest that dPDPN regulates the G2/M cell cycle in canine MM. One of the major regulators of the Chk and p53 cascade is ataxia telangiectasia (ATM), a DNA damage sensor. When cells receive DNA damage, ATM is phosphorylated by the meiotic recombination 11/DNA repair protein Rad50/Nijmegen breakage syndrome 1 protein (MRN) complex, which is the upstream of ATM, and the phosphorylated ATM activates downstream Chk and p53 cascades [73,74,75,76,77]. Previous reports have shown that, in canine MM cell lines, ATM is phosphorylated via endogenous DNA damage [78]. In this study, p-ATM expression was increased in the dPDPN-knocked-down Mi cell line compared to the control siRNA-treated cells. Therefore, it is possible that dPDPN suppresses the ATM signaling by interacting with the MRN complex, which, in turn, causes resistance to endogenous DNA damage. Previous research has shown that depletion of ROCK in mouse embryonic fibroblasts results in G2/M cell cycle arrest [79]. Thus, G2/M cell cycle arrest in dPDPN-knocked-down canine MM cell lines might be induced by the PDPN-induced inhibition of downstream ROCK signaling. Although the molecular mechanism by which dPDPN knockdown leads to G2/M cell cycle arrest was not clarified in this study, we have uncovered a novel role for PDPN in tumor cells. Further research is required to investigate the link between PDPN and cell cycle checkpoints.

In conclusion, our study demonstrated that 11 out of 18 types of canine tumors expressed dPDPN, and that dPDPN might be related to tumorigenesis and/or tumor malignancy in various types of canine tumors. Moreover, we showed, for the first time, that dPDPN knockdown suppressed cell viability by inducing apoptosis and G2/M cell cycle arrest in canine MM cell lines. Our findings indicate that dPDPN might play a crucial role in tumor progression in canine tumors with a high expression of dPDPN such as canine MM, SCC, and meningioma. In addition, these findings suggest the importance of dPDPN for clinical diagnosis and as a therapeutic target in these canine tumors. Furthermore, we believe that canine tumors expressing dPDPN could be used as models of their human counterparts.

## Figures and Tables

**Figure 1 cells-09-01136-f001:**
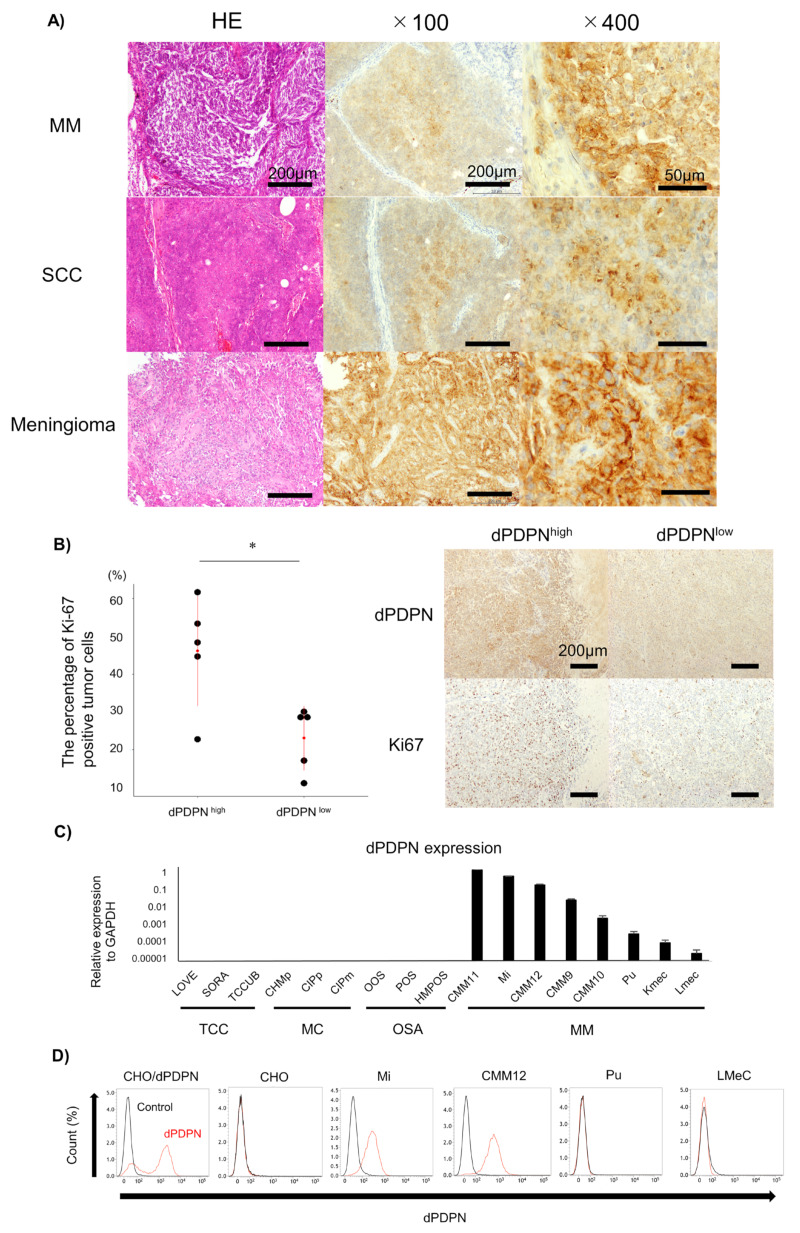
Evaluation of dPDPN expression in canine tumor tissues and canine cell lines. (**A**) Representative images of hematoxylin and eosin (HE) staining and immunohistochemistry of dPDPN in canine nine malignant melanoma (MM), squamous cell carcinoma (SCC) and meningioma are shown. Scale bar, left: 100 µm, center: 100 µm, right: 50 µm. (**B**) (Left panel) The percentage of Ki67-positive tumor cells in canine MM tissues was compared between the PDPN high-expression group and the PDPN low-expression group. Red point indicates the mean score and red bar indicates SD. Right panel: representative images of dPDPN and Ki67 expression in each group are shown. Scale bar: 200 µm. (**C**) dPDPN mRNA expression in various types of canine cell lines was evaluated by qPCR. Expression values were calculated relative to GAPDH expression. This experiment was performed in triplicate. (**D**) dPDPN protein expression in canine MM cell lines was evaluated by flow cytometry. Chinese hamster ovary (CHO)/dPDPN and CHO were used as positive and negative controls, respectively. * p < 0.05, hematoxylin and eosin (HE), malignant melanoma (MM), squamous cell carcinoma (SCC), dog podoplanin (dPDPN), transitional cell carcinoma (TCC), mammary carcinoma (MC), osteosarcoma (OSA), Chinese hamster ovary (CHO).

**Figure 2 cells-09-01136-f002:**
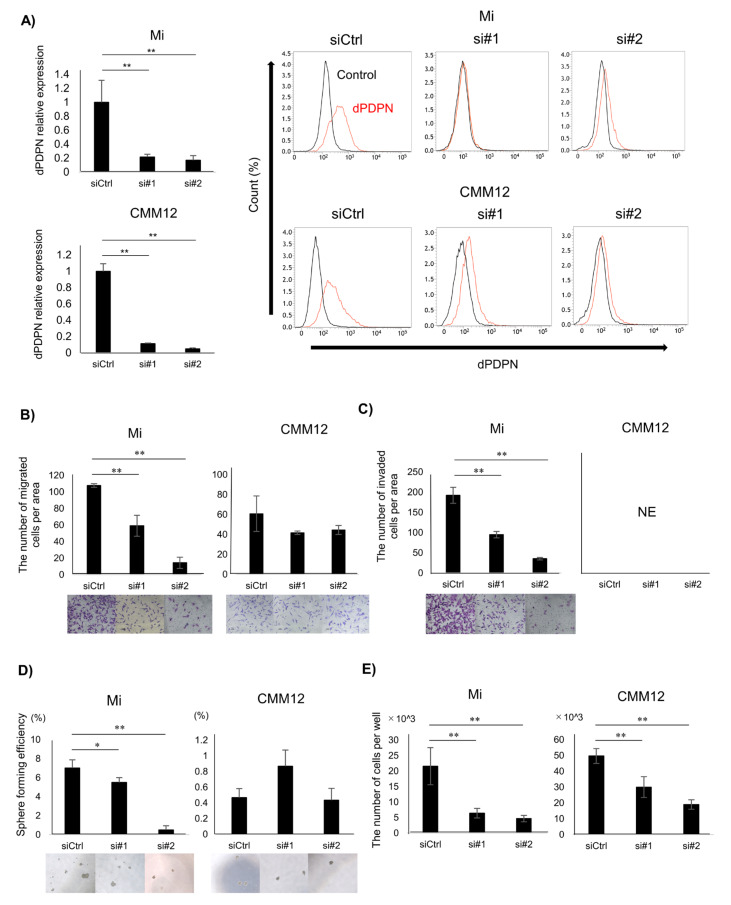
Investigating the association between dPDPN and tumor malignancy in canine MM cell lines. These experiments were performed in triplicate. (**A**) Left panel: *dPDPN* mRNA expression was evaluated by qPCR and (right panel) protein expression was evaluated by flow cytometry after 48 h of siRNAs treatment in Mi and CMM12 cell lines. (**B**) The number of migrated cells in control-scrambled siRNA, siRNA#1, or siRNA#2-treated Mi and CMM12 cell lines. Representative well images of each group are shown (×100). (**C**) The number of invading cells in control-scrambled siRNA, siRNA#1, or siRNA#2-treated Mi and CMM12 cell lines. Representative well images of each group are shown (×100). (**D**) Number of spheroids derived from control-scrambled siRNA, siRNA#1, or siRNA#2-treated Mi and CMM12 cell lines. Sphere forming efficiency is calculated by dividing the number of spheroids by the number of seeded cells. Representative well images of each group are shown (×100). (**E**) Number of living cells of control-scrambled siRNA, siRNA#1, or siRNA#2-treated Mi and CMM12 cell lines after 48 h of siRNAs treatment. siCtrl: Control-scrambled siRNA, si#1: siRNA#1, si#2: siRNA#2, * *p* < 0.05, ** *p* < 0.01, not evaluated (NE), dog podoplanin (dPDPN).

**Figure 3 cells-09-01136-f003:**
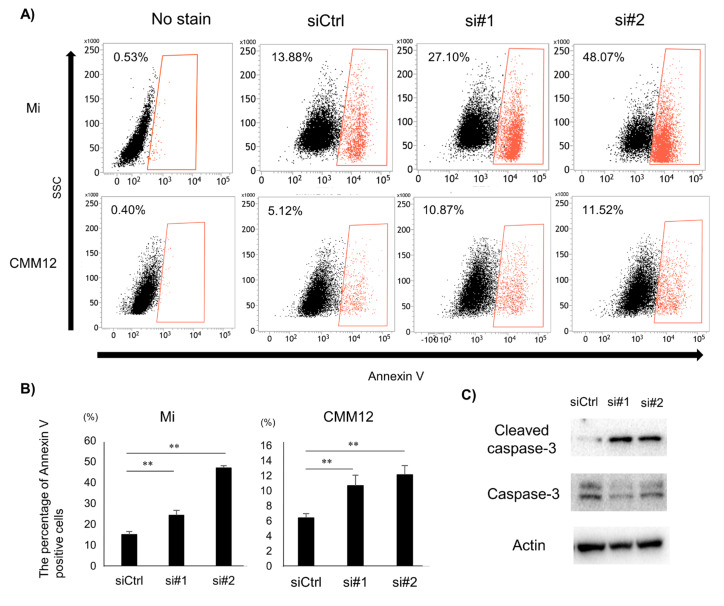
The effect of dPDPN knockdown on induction of apoptosis in canine MM. (**A**) Annexin V-positive cell populations after 48 h siRNA treatment were measured by flow cytometry. Representative images are shown. Non-stained cells were used as a control. (**B**) The percentage of Annexin V-positive cells in control-scrambled siRNA, siRNA#1, or siRNA#2-treated Mi and CMM12 cell lines. This experiment was performed in triplicate. (**C**) Western blot analysis of Cleaved Caspase 3 and Caspase 3 proteins after 48 h siRNA treatment for the Mi cell line. Actin was used as an internal control. **: *p* < 0.01, siCtrl: Control-scrambled siRNA, si#1: siRNA#1, si#2: siRNA#2.

**Figure 4 cells-09-01136-f004:**
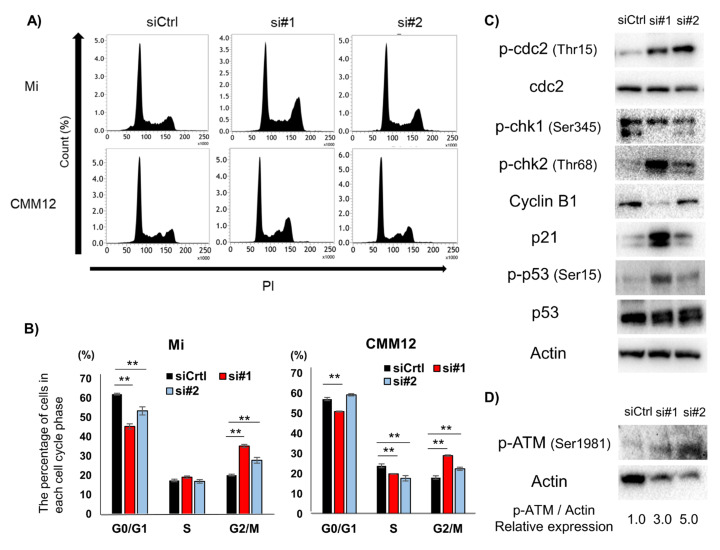
Effect of dPDPN knockdown on the cell cycle in canine MM cell lines. (**A**) Cell cycle analysis was performed in Mi and CMM12 cell lines after 48 and 72 h siRNA treatment, respectively. Representative images are shown. (**B**) The proportions of each cell cycle phase in Mi and CMM12 cell lines after siRNA treatment are summarized in graphs. This experiment was performed in triplicate. (**C**) G2/M cell cycle-related protein expression of Mi cell line after siRNAs treatment for 48 h was evaluated by Western blot analysis. Actin was used as an internal control. (**D**) The expression of phospho-ataxia telangiectasia-mutated (p-ATM) protein in the Mi cell line after siRNA treatment for 48 h was evaluated by Western blot analysis. The values indicated under each blot are the fold protein expression relative to the control after normalization by Actin (Image Lab quantification tool). **: *p* < 0.01, siCtrl: Control-scrambled siRNA, si#1: siRNA#1, si#2: siRNA#2, dog podoplanin (dPDPN), checkpoint kinase (Chk), ataxia telangiectasia-mutated (ATM).

**Table 1 cells-09-01136-t001:** Evaluation of dog podoplanin (dPDPN) expression in canine tumor tissues.

Tumor Types	Positive Ratio (%)
Malignant melanoma	8/10 (80%)
Squamous cell carcinoma	8/10 (80%)
Meningioma	4/5 (80%)
Lung adenocarcinoma	4/10 (40%)
Mast cell tumor	4/10 (40%)
Fibrosarcoma	3/10 (30%)
Prostate tumor	1/4 (25%)
Osteosarcoma	2/10 (20%)
Mammary carcinoma	2/10 (20%)
Peripheral nerve sheath tumor	1/10 (10%)
Anal sac adenocarcinoma	1/10 (10%)
Transitional cell carcinoma	0/10 (0%)
Thymoma	0/10 (0%)
Thyroid tumor	0/10 (0%)
Hepatic cell carcinoma	0/10 (0%)
Hemangiosarcoma	0/8 (0%)
Ceruminous carcinoma	0/8 (0%)
Intestinal adenocarcinoma	0/4 (0%)

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
