# Peer review of "PDPN Is Expressed in Various Types of Canine Tumors and Its Silencing Induces Apoptosis and Cell Cycle Arrest in Canine Malignant Melanoma"

_cells, 2020, doi:10.3390/cells9051136_

Round 1

Reviewer 1 Report

In this manuscript, Shinada at al report that PDPN expression is increased in many types of canine cancers, particularly malignant melanoma. They also report that PDPN appears to promote malignant melanoma cell proliferation as evidenced by ki67 expression. In addition, they find that PDPN also promotes melanoma cell migration, stemness, and viability. They also present findings that suggest specific pathways by which PDPN acts to promote tumor progression.

This is a robust study. The experiments are done with pertinent controls, performed in a logical progression, and the data are clearly interpreted. This report is bound to be very well appreciated by the research community, and serve to assist other investigators perform meaningful experiments. There are just some minor concerns that should be addressed.

References numbers #62 and #76 appear to be incomplete.

The authors may consider referencing Ochoa-Avlarez et al (PLoS One 2012) as evidence of PDPN expression in malignant melanoma supported along with reference #31 on page 10 (line 323).

The authors should consider adding lanes to show effects of siRNA on PDPN expression in Western blots shown in Figure 3B and Figure 4C.

Reviewer 2 Report

In summary, it's an interesting study whose results for canine tumors could be also valuable for human cancer mechanisms. But the form of the manuscript has to be improved in several points.

Introduction is a little bit short. Parts of the Discussion section describing frequencies and mortality would have fit here.

Material & Methods:
IHC: the technical part (2.1) is written too extensive, but in the evaluation part (2.2), a clear algorithm is missing, e.g. is the staining intensity also taken into account;
qPCR(2.5): protocol and PCR conditions are missing; how many replicates?;

siRNA (2.7): RNA sequences (also siRNA sequences) use "U" instead of "T"

Transwell assays (2.8): definition of "area", e.g. in square mm

Results and Figures:

Figures in general: bad qualtiy, not well readable; IHCs too small; labeling blurred; must be improved

Specific remarks:

1B: missing definition of asteriks (significance)

1C: missing error bars

Reviewer 3 Report

This manuscript entitled “PDPN is expressed in various types of canine tumors 2 and its silencing induces apoptosis and cell cycle 3 arrest in canine malignant melanoma” by Masahiro Shinada et al. demonstrated dPDPN expression in various types of canine tumors and investigated the function of dPDPN in canine malignant melanoma cell lines. This is novel to investigate dPDPN in canine malignancies but the experiments performed were not solid enough to prove the concept and mechanism.

Major comments:

  1. Annexin V in Figure 3A. According to description and figures, I am not sure how many repeats done for this experiment. I suggest at least 3 repeats for scientific study and present it with mean + SD with error bar in another panel in this figure.
  2. The histograms of cell cycle (Figure 4A) were not well performed as a high peak of sub-G1 appeared in the control indicating that the cells were not in good condition for experiments. The authors should optimize the experimental condition for cell cycles. In addition, the sub-G1 phase seems decreased after siRNA knockdown implying siRNA reduced apoptosis which is not consistent with the results from figure 3. The authors should explain this.
  3. The authors did not show the mechanism how dPDPN regulate apoptosis and cell cycle arrest. In discussion, the authors mentioned one of the major regulators of Chk and p53 cascade is ataxia telangiectasia (ATM). In addition, ATM is the major kinase which can phosphorylate p53 at serine 15. Therefore, the authors should investigate the changes of ATM and pATM or use ATM inhibitor to see if this finding is ATM-dependent.

Minor comments:

  1. The figure 1 is not clear enough to review
  2. The authors should list the full words of abbreviation in the legends of the figures.
  3. Line 257: The same was not observed in the CMM12 cell line (Figure 2D). à the same ?
  4. Figure 2D : Sphere forming assay for CMM12 illustrated only few sphere formation for the control and I suppose this might possibly result from fewer cells seeded or shorter incubation time. The author should seed more cells for this assay as CMM12 grows much slower and is less invasive than Mi.
  5. The authors did not mention how long for siRNA incubation for sphere forming assay.
  6. Line 167 Cells were incubated with siRNAs for 48-72 h. What is the exact incubation time for cell cycle ? I could not find this in the manuscript.
  7. Figure 4C: no total chk1, chk2 were found in immunoblotting. In addition, the authors should show the expression of dPDPN which is the main target in current study.
  8. Line 323 : human skin cutaneous MM à skin = cutaneous
  9. SCC can occur in various organs in the body such as head and neck, lung, skin, and etc. The authors did not specify the origin of SCC.
  10. Line 326 : Canine MM is the most common oral malignancy in dogs. The oral melanoma is classified as mucosal melanoma and is much different from cutaneous melanoma. The authors should discuss them separately.
  11. Line 442: dPDPN acts by inhibiting ATM phosphorylation à how ? is it a phosphatase ?

Round 2

Reviewer 3 Report

This manuscript has been great improved by authors. I consider this could be published after minor revision.

  1. The authors mentioned "We previously tried to detect dPDPN by Western blot analysis using our antibody, but we could not detect dPDPN. Therefore, we used flowcytometric analysis to detect dPDPN protein expression in this research." I suggest the authors should put this in the limitation as the readers will be interested in this.
  2. The authors added “All canine MM tissues were derived from oral cavity, and canine SCC tissues were derived from different organs; tonsil (n=4), lip (n=2), mandibular (n=2), head and neck (n=1) and skin (n=1).”.  "Mandibular" is an adj. indicating mandible which is a bone of jaw. I don't consider SCC would be derived from this. In addition, tonsil, lip, mandible, are classified as head and neck cancer so the authors could combined these together or describe separately.

  3. “it is possible that dPDPN inhibit ATM phosphorylation by regulating upstream of ATM” --> authors could add more details about this. In the reply form, they mention "(e.g. dPDPN suppresses ATM signalling by interacting with MRN proteins complex.)" which could be added in the discussion and cite the reference.
